# GAN-argcPredNet v1.0:A Generative Adversarial Model for Radar Echo Extrapolation Based on Convolutional Recurrent Units

Kun Zheng[1], Yan Liu[1], Jinbiao Zhang[2], Cong Luo[3], Siyu Tang[3], Huihua Ruan[2], Qiya Tan[1], Yunlei Yi[4], Xiutao Ran[4]

[1]School of Geography and Information Engineering, China University of Geosciences, Wuhan 430074, China
[2]Guangdong Meteorological Observation Data center, Guangzhou 510080, China
[3]GuangDong Meteorological Observatory, GuangDong, 510080, China
[4]Wuhan Zhaotu Technology Co. Ltd., Wuhan 430074, China
*Correspondence to*: Kun Zheng (ZhengK@cug.edu.cn);Yan Liu(liuyan_@cug.edu.cn)

**Abstract.** Precipitation nowcasting play a vital role in preventing meteorological disasters and doppler radar data acts as an important input for nowcasting models. The traditional extrapolation method is difficult to model highly nonlinear echo movements. The key challenge of the nowcasting mission lies in achieving high-precision radar echo extrapolation. In recent years, machine learning has made a great progress in the extrapolation of weather radar echoes. However, most of models neglect the multi-modal characteristics of radar echo data, resulting in blurred and unrealistic prediction images. This paper aims to solve this problem by utilizing the feature of the GAN that can enhance the multi-modal distribution modelling, and design the radar echo extrapolation model of GAN-argcPredNet v1.0. Model is composed of argcPredNet generator and a convolutional neural network discriminator. In the generator, a gate controlling the memory and out are designed in the rgcLSTM component, thereby reducing the loss of spatiotemporal information. In the discriminator, the model uses a dual-channel input method, which enables it to strictly score according to the true echo distribution, and has a more powerful discrimination ability. Through experiments on the radar data set of Shenzhen, China, the results show that the radar echo hit rate (POD) and critical success index (CSI) have an average increase of 21.4% and 19% compared with rgcPredNet under different intensity rainfall thresholds, and the false alarm rate (FAR) has decreased by an average of 17.9%. From the comparison of the result graph and the evaluation index, we also found a problem. The recursive prediction method will produce the phenomenon that the prediction result will gradually deviate from the true value over time. In addition, the accuracy of high-intensity echo extrapolation is relatively low. This is a question worthy of further investigation. In the future, we will continue to conduct research from these two directions.

## 1 Introduction

Precipitation nowcasting refers to the prediction and analysis of rainfall in the target area in a short period of time (0-6 hours) (Bihlo et al. 2019, Luo et al. 2020). The important data needed for this work comes from doppler weather radar with high temporal and spatial resolution (Wang et al. 2007). Relevant departments can issue early warning information through accurate nowcasting to avoid loss of economic life (C et al. 2021). However, this task is extremely challenging due to its very low tolerance to time and position errors (Sun et al. 2014).

The existing nowcasting systems mainly include two types, numerical weather prediction (NWP) and based on radar echo extrapolation (Chen et al. 2020). The widely used optical flow method has problems such as poor capture of fast echo change regions, high complexity of the algorithm and low efficiency (Shang et al. 2017). Since echo extrapolation can be considered as a time series image prediction problem, these shortcomings of optical flow method are expected to be solved by recurrent neural network (RNN) (Giles et al. 1994).

With the continuous development of deep learning, more and more neural networks have been applied to the field of nowcasting. Forecast models such as ConvLSTM and EBGAN-Forecaster show that its extrapolation effect is better than that of optical flow method (Shi et al. 2015, Chen et al.2019). However, these models still have the problem of blurred and

unrealistic prediction images (Tian et al. 2020, Xie et al. 2020, Jing et al. 2019). One of the main reasons is that radar echo maps are typical multi-modal data acquired by multiple sensors and different stations, some algorithms ignore this feature of radar echo maps, using the mean square error and mean absolute error as the loss function, which is better suited to a unimodal distribution.

The paper proposes a GAN-argcPredNet network model, which aims to solve this problem through GAN's ability to strengthen the characteristics of multi-modal data modelling. The generator adopts the same deep coding-decoding method as PredNet to establish a prediction model, and uses a new structure of convolutional LSTM as a predictive neuron, which can effectively reduce the loss of spatiotemporal information compared with rgcLSTM. The deep convolutional network is used as the discriminator to classify, and the dual-channel input mechanism is used to strictly judge the distribution of real radar

echo images. Finally, based on the weather radar echo data set, the generator and the discriminator are alternately trained to make the extrapolated radar echo map more real and precise.

## 2 Related work

### 2.1 Sequence prediction networks

The essence of radar echo image extrapolation is the problem of sequence image prediction, which can be solved by

implementing an end-to-end sequence learning method (Shi et al. 2015, Sutskever et al.2014). ConvLSTM introduces a convolution operation in the conversion of the internal data state of the LSTM, effectively utilizing the spatial information of the radar echo data (Shi et al. 2015). However, because the location-invariant of the convolutional recursive structure is inconsistent with the natural change motion, TrajGRU was further proposed (Shi et al. 2017). GRU (Gated Recurrent Unit), as a kind of recurrent neural network, it performs to LSTM but is computationally cheaper (Group et al. 2017). Similarly,

ConvGRU introduces convolution operations inside the GRU to enhance the sparse connectivity of the model unit and is used to learn video spatiotemporal features (Ballas et.al 2015). The RainNet network learns the movement and evolution of radar echo based on the U-NET convolutional network for extrapolation prediction (Ayzel et al. 2020). PredNet is based on a deep coding framework and adds error units to each network layer, which can transmit error signals like the human brain structure (Lotter et.al 2016). In order to increase the depth of the network and the connections between modules, Skip-PredNet further

introduces skip connections and uses ConvGRU as the core prediction unit. Experiments show that its effect is better than the TraijGRU benchmark (Sato et.al 2018). Although these networks can achieve echo prediction, they have the problem of blurring and unrealistic extrapolated images.

### 2.2 GAN-based radar echo extrapolation

The Generative Adversarial Network (GAN) consists of two parts: a generator and a discriminator (Goodfellow et al. 2014).

GAN can be an effective model for generating images. Using an additional GAN loss, model can better achieve multi-modal data modelling, and each of its outputs is clearer and more realistic (Lotter et.al 2016). Multiple complementary feature learning strategies show that generative adversarial training can maintain the sharpness of future frames and solve the problem of lack of clarity in prediction (Michael et.al 2015). In this regard, the extrapolators built a generative adversarial network to solve the problem of extrapolated image blur, trying to use this adversarial training to extrapolate more detailed radar echo maps (Singh

et al. 2017). Similarly, adversarial network with ConvGRU as the core was proposed, mainly to solve the problem of ConvGRU's inability to achieve multi-modal data modelling (Tian et.al 2020). There are also researchers based on the idea of a four-level pyramid convolution structure, and proposed four pairs of models to generate an adversarial network for radar echo prediction (Chen et al. 2019). It should be noted that the traditional GAN network has the problem of unstable training, which will cause the model unable to learn. Therefore, the design of the nowcasting model should be based on a stable and

optimized GAN network.

## 3 Model

In this section, we describe the model from the overall to the details. Section 3.1 introduces the overall structure and training process of the model. In section 3.2, we describe the structure of the argcPredNet generator and focus on the argcLSTM neuron. In section 3.3, the paper introduces the design of the discriminator and the loss function of the model.

### 3.1 GAN-argcPredNet model overview

Radar echo extrapolation refers to the prediction of the dissipation and distribution of future echoes based on the existing radar echo sequence diagram. If the problem is formulated, then each echo maps can be regarded as a tensor $x \in R^{W \times H \times C}$, W、 H、 C represent the width, height, and number of channels, respectively, and R represents observing the feature area. If input M sequence echo maps, predict the N most likely changes in the future, this problem can be expressed in Eq. (1). This article sets the input sequence M and output sequence N to 5 and 7, respectively.

$$\hat{x}^{t+1},..., \ \hat{x}^{t+N} = \mathop{\mathrm{argmax}}_{x^{t+1},..,x^{t+N}} p(x^{t+1},..., \ x^{t+N}|x^{t-M+1},...,x^{t}) \tag{1}$$

Unlike other forecasting models, GAN-argcPredNet uses WGAN-gp (Wasserstein Generative Adversarial Network with Gradient penalty) as a predictive framework. The model solves the problem of training instability through gradient penalty measures (Gulrajani et al.2017). Our model mainly includes two parts: generator and discriminator. As shown in Fig.1, the generator is composed of argcPredNet, which is responsible for learning the potential features of the data and simulating the data distribution to generate prediction samples. Then, the predicted samples and the real samples are input into the discriminator to make a judgment, the real data is judged to be true, and the predicted data is judged to be false. Finally, we use the Adam optimiser for training the adversarial loss and then update the parameters of the discriminator, optimize the loss function of the generator once every 5 updates, and complete the update of the generator parameters. The algorithm flow is shown in Table 1(Gulrajani et.al 2017).

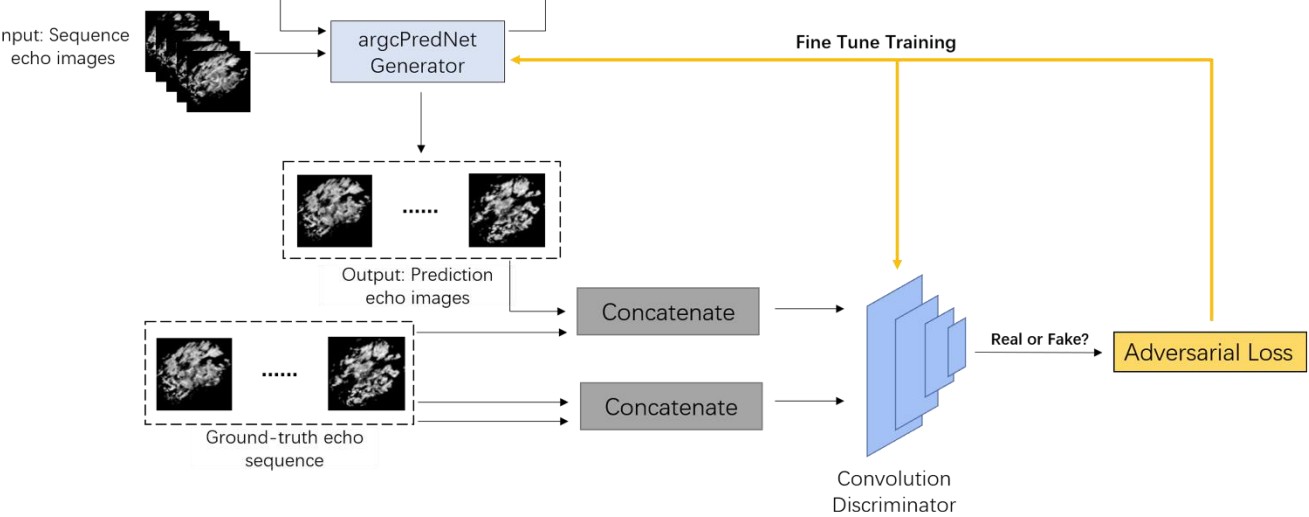

**Figure 1. The schematic of the GAN-argcPredNet architecture.**

Table.1 GAN-argcPredNet training algorithm flow

| **Algorithm** Model uses default values of $\lambda = 10$, $n_{critic} = 5$, $\alpha = 0.0001$, $\beta_1 = 0.5$, $\beta_2 = 0.9$. |
|---|
| Parameters: The gradient penalty coefficient λ, the number of critic iterations per generator iteration $n_{critic}$, the batch size $m$, Adam hyperparameters $\alpha$, $\beta_1$, $\beta_2$, initial critic parameters $\omega_0$, initial generator parameters $\theta_0$. |
| 1. **for** $i = 1, \cdots, epoch$ **do** |
| 2.     **for** $t = 1, \cdots, n_{critic}$ **do** |

| | |
|---|---|
| 3. | **for** $j = 1, \cdots, m$ **do** |
| 4. | Sample real data $x \sim P_r$, latent variable $z \sim p_z$, a random number $\epsilon \sim \cup [0,1]$ |
| 5. | $\tilde{x} \leftarrow G_\theta(z)$ |
| 6. | $\hat{x} \leftarrow \epsilon x + (1 - \epsilon)\tilde{x}$ |
| 7. | $L^{(j)} = D_\omega(\tilde{x}) - D_\omega(x) + \lambda(\|\nabla_{\hat{x}} D_\omega(\hat{x})\|_2 - 1)^2$ |
| **8.** | **end for** |
| 9. | $\omega \leftarrow Adam(\nabla_\omega \frac{1}{m} \sum_{j=1}^{m} L^{(j)}, \omega, \alpha, \beta_1, \beta_2)$ |
| **10.** | **end for** |
| 11. | Sample a batch of latent variables $\{z^j\}_{j=1}^{m} \sim P_z$s |
| 12. | $\theta \leftarrow Adam(\nabla_\theta \frac{1}{m} \sum_{j=1}^{m} - D_\omega(G_\theta(z)), \theta, \alpha, \beta_1, \beta_2)$ |
| **13. end for** | |

### 3.2 argcPredNet generator

#### 3.2.1 argcLSTM

The internal structure of the argcLSTM neuron used in the model is shown in Fig.2. In order to provide better feature extraction capabilities, the structure contains two trainable gating units, one is the forget gate $f^{(t)}$, the other is the input gate $g^{(t)}$. The latter can calculate the weight of the current state independently, and complete the feature retention of the input information. The peephole connection from the unit state to the forget gate is removed. This operation does not have a big impact on the
result, but simplifies the redundant parameters. The complete definition of the argcLSTM unit is as follows (Eq. (2) – Eq. (6)).

$$f^{(t)} = \sigma\left(W_{fx} * x^{(t)} + U_{fh} * h^{(t-1)} + b_f\right) \tag{2}$$

$$g^{(t)} = \sigma\left(W_{gx} * x^{(t)} + U_{gh} * h^{(t-1)} + b_g\right) \tag{3}$$

$$\tilde{C}^{(t)} = \tanh\left(W_{ch} * h^{(t-1)} + W_{cx} * x^{(t)} + b_c\right) \tag{4}$$

$$C^{(t)} = f^{(t)} \circ C^{(t-1)} + g^{(t)} \circ \tilde{C}^{(t)} \tag{5}$$

$$h^{(t)} = g^{(t)} \circ \tanh\left(C^{(t)}\right) \tag{6}$$

Among them, $*$ represents convolution operation, $\circ$ represents Hadamard product, $\sigma$ represents sigmoid nonlinear activation function, $f^{(t)}$、$g^{(t)}$ represent forget gate and update gate, $x^{(t)}$、$h^{(t)}$、$C^{(t)}$ represent the input, hidden state and unit state at time $t$, respectively.

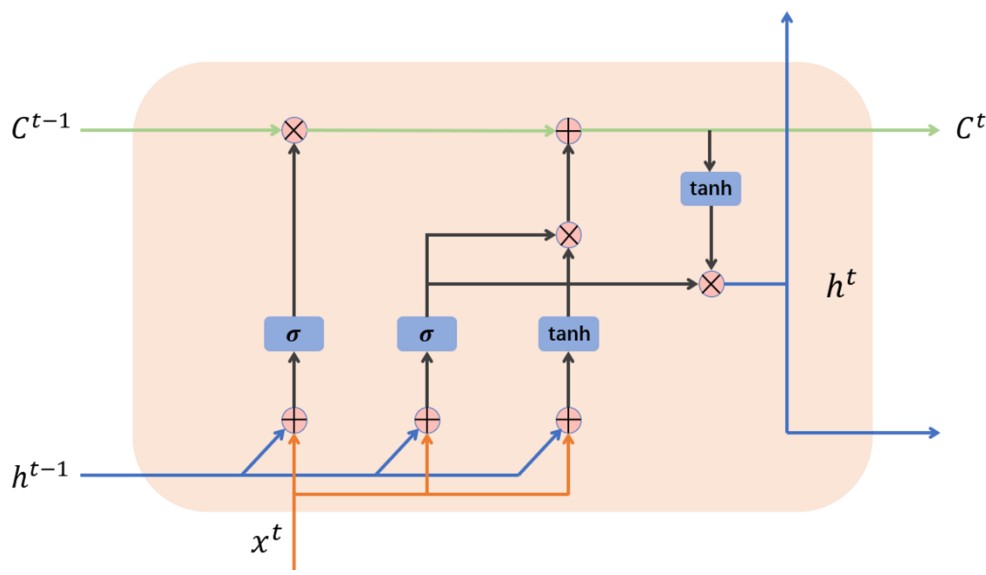

**Figure 2. argcLSTM internal structure**

### 3.2.2 argcPredNet

The argcPredNet generator has the same structure as PredNet, which is composed of a series of repeatedly stacked modules, stacking a total of 3 layers. The difference is that argcPredNet uses argcLSTM as the prediction unit. As shown in Fig 3, each layer of the module contains four units, namely: $A_l$: input convolutional layer, $R_l$: recurrent representation layer, $\hat{A}_l$: prediction convolutional layer, $E_l$: error representation layer.

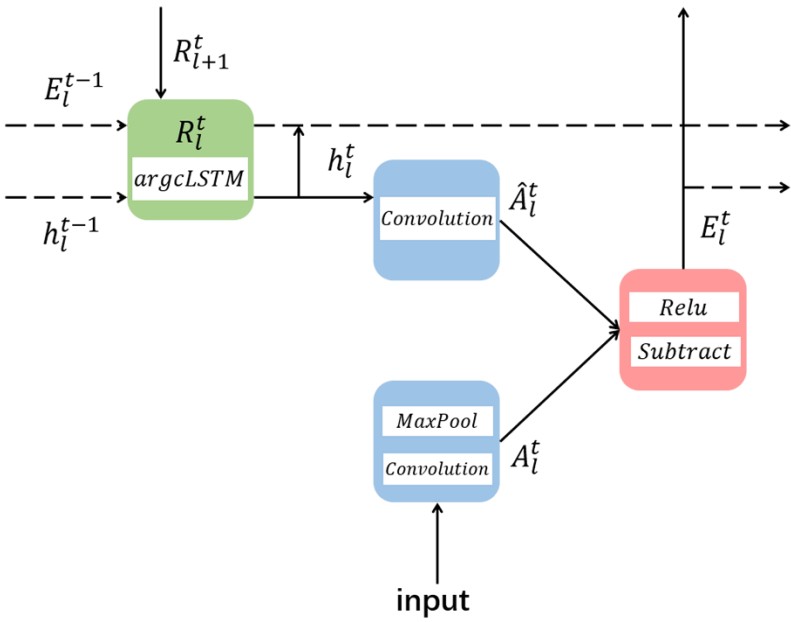

**Figure 3. Module expansion diagram of layer $l$ at time $t$**

The recursive prediction layer uses the argcLSTM loop unit, which is used to generate the prediction of the next frame and the input of $\hat{A}_l$ and $A_{l+1}$. The network uses error calculation, $E_l$ will output an error representation, and then the error representation is passed forward through the convolutional layer to become the input of the next layer $A_{l+1}$. The hidden state of the recurrent unit $R_l^t$ is updated according to the output of $E_l^{t-1}$, $R_l^{t-1}$ and the up-sampled $R_{l+1}^t$. For $A_l$, the input of the lowest target, namely $A_0$, is set to the actual sequence itself, when $l > 0$, the input of $A_l$ is: lower error signal $E_{l-1}$ results from convolution calculation, RELU activation and maximum pooling layer. The complete update rules are shown in Eq. (7) to Eq. (10). The specific parameter settings of the generator are shown in Table 2. (1, 128, 128, 256) represents the number of filters from the first layer to the fourth layer from left to right.

$$A_l^t = \begin{cases} x_t & if \ l = 0 \\ MAXPOOL\left(RELU\left(CONV\left(E_{l-1}^t\right)\right)\right) & 0 < l < L \end{cases} \tag{7}$$

$$\hat{A}_l^t = RELU\left(CONV\left(R_l^t\right)\right) \tag{8}$$

$$E_l^t = \left[RELU\left(A_l^t - \hat{A}_l^t\right) ; \ RELU\left(\hat{A}_l^t - A_l^t\right)\right] \tag{9}$$

$$R_l^t = \begin{cases} argcLSTM\left(E_l^{t-1}, \ R_l^{t-1}\right) & if \ l = L \\ argcLSTM\left(E_l^{t-1}, \ R_l^{t-1}, \ UPSAMPLE\left(R_{l+1}^t\right)\right) & 0 < l < L \end{cases} \tag{10}$$

Table 2. Generator parameter settings

| Components | Name | Filter size | Filter Numbers |
|---|---|---|---|
| Module A | Convolution layer | 3 x 3 | (1, 128, 128, 256) |

| | | | |
|---|---|---|---|
| | Max pool | 3 x 3 | / |
| Unit $\hat{A}_l^t$ | Convolution layer | 3 x 3 | (1, 128, 128, 256) |
| Module R | Up sample | 3 x 3 | / |
| | argc_LSTM | 3 x 3 | (1, 128, 128, 256) |

## 3.3 Discriminator and loss

### 3.3.1 Convolutional discriminator

The purpose of the discriminator is to recognize images, which is similar in nature to the classifier. In the GAN-argcPredNet model, a DC-CNN network is designed for discrimination. The process is shown in Fig.4. It is a four-layer convolution model with a dual-channel input method.

The DC-CNN network extracts a pair of images from the three pairs of images, and inputs them to the fully connected layer through a four-layer convolution transformation, and finally generates a probability output through the Sigmoid function, indicating the possibility that the input image is from a real image. When the input is a real image, the discriminator will maximize the probability, and the value will approach 1. If the input is a generator synthesized image, the discriminator will minimize the probability, and the value will approach -1. The specific parameter settings of the discriminator are shown in Table 3.

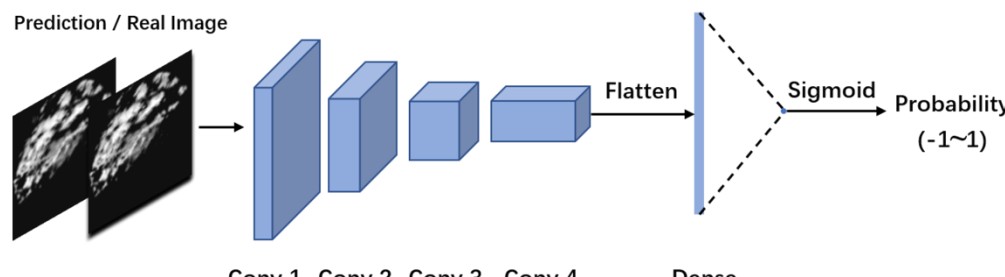

**Figure 4. DC-CNN structure**

Table 3. Discriminator parameter settings

| Name | Filter size | Stride | Filter numbers | Output size |
|---|---|---|---|---|
| Convolution_1 | 3 x 3 | 2 x 2 | 32 | 48 x 48 |
| Convolution_2 | 3 x 3 | 2 x 2 | 64 | 24 x 24 |
| Convolution_3 | 3 x 3 | 2 x 2 | 128 | 12 x 12 |
| Convolution_4 | 3 x 3 | 2 x 2 | 256 | 6 x 6 |

### 3.3.2 Loss function

The generative adversarial network relies on the distribution of simulated data to generate images. It can retain more echo details, thereby realizing the modelling of multi-modal data. A gradient penalty term is added to GAN-argcPredNet, the loss function of the discriminator is shown in Eq. (11).

$$L_D = D(\tilde{x}) - D(x) + \lambda(\|\nabla_{\hat{x}} D(\hat{x})\|_2 - 1)^2 \tag{11}$$

The generator has the following loss function (Eq. (12)).

$$L_G = E_{\tilde{x} \sim P_g}[D(\tilde{x})] - E_{x \sim P_r}[D(x)] \tag{12}$$

The model has the following maximum-minimum loss function (Eq. (13)).

$$\min_{G} \max_{D} V(D, G) = E_{\tilde{x} \sim P_g}[D(\tilde{x})] - E_{x \sim P_r}[D(x)] + \lambda E_{\hat{x} \sim P_{\hat{x}}}(\|\nabla_{\hat{x}} D(\hat{x})\|_2 - 1)^2 \tag{13}$$

Where $\tilde{x}$ represents the distribution of generated samples. $P_g$ represents the set of generated sample distributions. $x$ represents the distribution of real samples. $P_r$ represents the set of real sample distributions. The third term is the penalty item of the gradient penalty mechanism. In the penalty term, $\hat{x}$ represents the actual data and generation a new sample formed by random sampling between data. $P_{\hat{x}}$ represents a set of randomly sampled samples. $\lambda$ is a hyperparameter, which represents the coefficient of the penalty term, and the value in the model is set to 10.

## 4 Experiments

In order to verify the effectiveness of the model, the paper uses the radar echo data from January to July 2020 in Shenzhen, China, to conduct experiments on the four models of ConvGRU, rgcPredNet, argcPredNet and GAN-argcPredNet. All experiments are implemented in Python, based on the Keras deep learning library, with Tensorflow as the backend for model training and testing.

### 4.1 Dataset description

This experiment uses the radar echo data of Shenzhen China. The data set is all rain images after quality control. The reflectivity range is 0-80dBZ, the amplitude limit is between 0 and 255, and it is collected every 6 minutes, with a total of 1 layer. The height of sea level is 3km. A total of 600,000 echo images were collected, of which 550,000 were used as the training set and 50,000 were used as the test set for testing. Each set of data contained 12 consecutive images. The horizontal resolution of the radar echo maps is 0.01 degrees (about 1km), the number of grids is 501*501 (that is, an area of about 500km×500km), and the image resolution is 96*96 pixels.

### 4.2 Evaluation metrics

In order to evaluate the accuracy of the model on precipitation nowcasting, the experiment uses three evaluation indicators to evaluate the prediction precision of the model, critical success index (Eq. (14)), namely false alarm rate (Eq. (15)) and hit rate (Eq. (16)).

$$CSI = \frac{TP}{TP+FN+FP} \tag{14}$$

$$FAR = \frac{FN}{TP+FN} \tag{15}$$

$$POD = \frac{TP}{TP+FP} \tag{16}$$

In the formula, TP indicates that both the predicted value and the true value reach the specified threshold, FN means that the true value reaches the specified threshold but the predicted value has not reached, FP indicates that the true value has not reached the specified threshold but the predicted value has reached the specified threshold.

### 4.3 Results

The experiment comprehensively evaluates the prediction accuracy of precipitation with different thresholds. The radar reflectivity and rainfall intensity refer to the Z-R relationship (Shi et al. 2017). The calculation formula is as Eq. (17).

$$Z = 10 \log a + 10b \log R \tag{17}$$

In this paper, $a$ is set to 58.53, and $b$ is set to 1.56, Z represents the intensity of radar reflectivity, $R$ represents the intensity of rainfall, and the corresponding relationship between rainfall and rainfall level refers to Table 4 (Shi et al .2017).

Table 4. Rain level

| Rain Rate(mm $h^{-1}$) | Rainfall Level |
|---|---|
| $0 \leq R < 0.5$ | No / Hardly noticeable |
| $0.5 \leq R < 2$ | Light |
| $2 \leq R < 5$ | Light to moderate |
| $5 \leq R < 10$ | Moderate |
| $10 \leq R < 30$ | Moderate to heavy |
| $30 \leq R$ | Rainstorm warning |

Figure 5, Figure 6 and Figure 7 compare the CSI, POD, and FAR index scores of each model at different rainfall thresholds in detail.

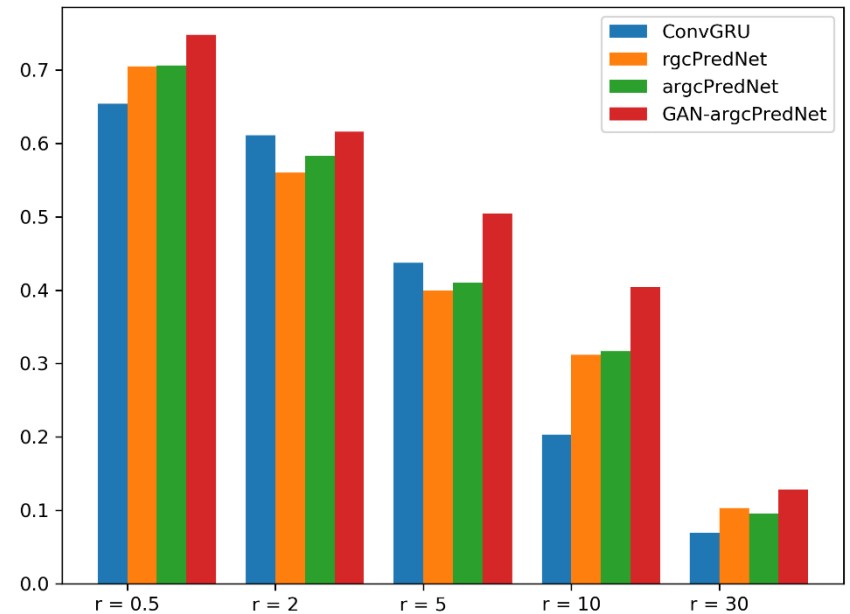

**Figure 5.CSI Index score**

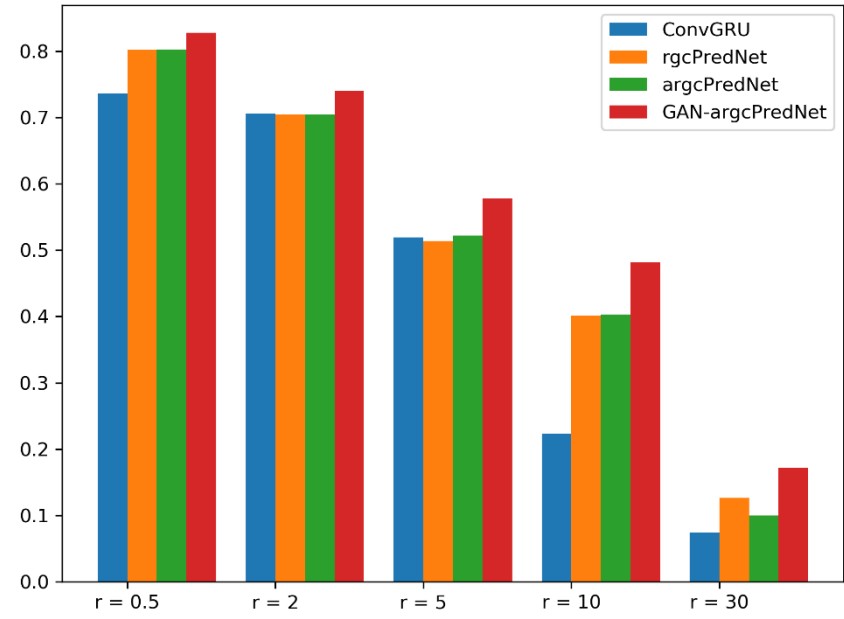

**Figure 6. POD Index score**

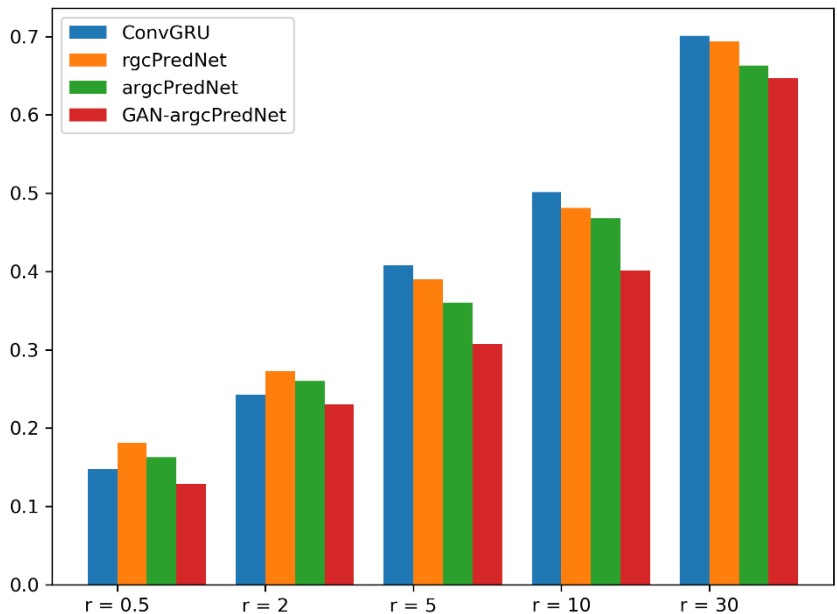

**Figure 7. FAR Index score**

This result is calculated based on 50,000 test pictures (approximately more than 4,000 test sets), which is representative. It can be seen that when the rainfall rate increases from 0.5 mm $h^{-1}$ to 30 mm $h^{-1}$, GAN-argcPredNet always performs best, and its advantage is very significant, argcPredNet is the second, and the ConvGRU model is the worst. Another point worth noting is that as the rainfall intensity increases, the performance of all models shows a downward trend. In the comparison of CSI indicators, GAN-argcPredNet leads the rest of the models when the rainfall rate is lower than when the rainfall rate is lower than 30 mm $h^{-1}$. When the rainfall level is rainstorm warning, its leading advantage is the weakest. And argcPredNet only leads rgcPredNet by a slight advantage, and its performance with rgcPredNet is not as good as ConvGRU in the range of 2-5 mm $h^{-1}$. For POD indicators, GAN-argcPredNet performs best, and its leading advantage is more prominent. The performance of argcPredNet is not so outstanding, almost the same as argcPredNet, but the index of the two is always better than ConvGRU. For the FAR score, the performance of GAN-argcPredNet is still the best, while argcPredNet and rgcPredNet are worse than ConvGRU in the range of 2-5 mm $h^{-1}$, and the score in the interval of rain rate 0.5-10 mm $h^{-1}$is slightly lower than that of GAN-argcPredNet.

To compare the three methods more intuitively, Figure 8 show the image prediction results of the three models on the same piece of test data.

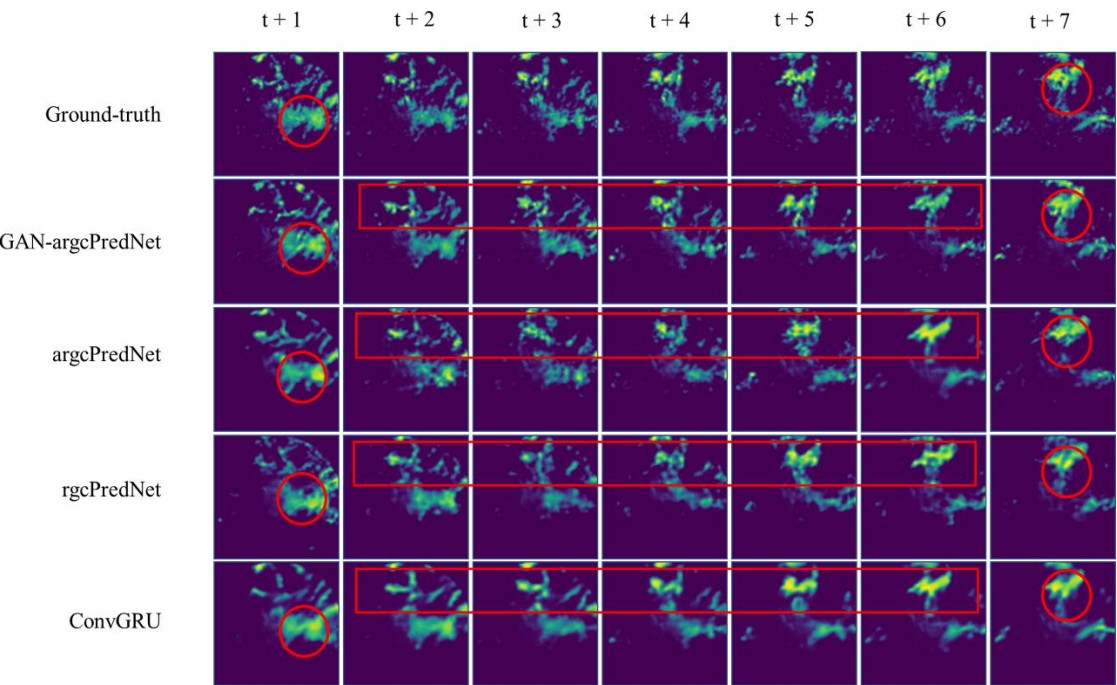

**Figure 8. Four prediction examples for the precipitation nowcasting problem. From top to bottom: ground truth frames, prediction by GAN-argcPredNet, prediction by argcPredNet, prediction by rgcPredNet, prediction by ConvGRU**

Compared with the other three models, GAN-argcPredNet generates better image clarity and shows more detailed features on a small scale. The contrast between the areas marked by the red ellipse in Fig. 8 is more obvious. GAN-argcPredNet has made the best prediction on the shape and intensity of the echo. The area selected by the rectangle mainly shows the echo changes in the northern region within 30 minutes. Both models correctly predict the movement of the echo to a certain extent, and the prediction process shown by GAN-argcPredNet is the most complete. In some mixed intensity and edge areas, our
model clearly predicts the echo intensity information, which can be seen the effect of confrontation training is obvious.

In order to compare the prediction results more specifically, the experiment uses Mean Square Error (MSE) and Mean Structural Similarity (MSSIM) to evaluate the quality of the generated images (Wang et al. 2004). The MSE and MSSIM index scores of the images generated by each model are shown in Table 5. ConvGRU has the lowest two indexes. Although the MSE index of rgcPredNet is slightly lower than that of the argcPredNet and GAN-argcPredNet models, the MSSIM index of the
argcPredNet and GAN-argcPredNet models is 0.066 and 0.109 higher than that of the rgcPredNet network model, respectively.

Table 5. MSE and MSSIM index scores of each model

| Name | MSE $\times 10^2$ ↓ | MSSIM ↑ |
|------|------|------|
| ConvGRU | 0.950 | 0.705 |
| rgcPredNet | 0.496 | 0.724 |
| argcPredNet | 0.476 | 0.790 |
| GAN-argcPredNet | **0.451** | **0.833** |

## 5 Conclusion

The study demonstrated a radar echo extrapolation model. The main innovations are summarized as follows. First, the argcPredNet generator is established based on the time and space characteristics of radar data. argcPredNet can predict future
echo changes based on historical echo observations. Second, our model uses adversarial training methods to try to solve the problem of blurry predictions.

From the evaluation indicators and qualitative analysis results, GAN-argcPredNet has achieved excellent results. Our model can reduce the prediction loss in a small-scale space, so that the prediction results have more detailed features. However, the recursive extrapolation method causes the error to accumulate as time goes by, and the prediction result deviates more and

more from the true value. In addition, when the amount of high-intensity echo data is small, the prediction of high-risk and strong convective weather through machine learning is also a problem that we are very concerned about, because it is more realistic. So, we will carry out research on these two issues in the future.

Code and data availability. The GAN-argcPredNet and argcPredNet models are free and open source. The current version number is GAN-argcPredNet v1.0, and the source code is provided through a GitHub repository https://github.com/Luka-
Doncic0/GAN-argcPredNet or accessed through the zendo repository https://doi.org/10.5281/zenodo.5035201. The pretrained GAN-argcPredNet and argcPredNet weights are available at https://doi.org/10.5281/zenodo.4765575. The radar data used in the article comes from the Guangdong Meteorological Department. Due to the confidentiality policy, the data will not be disclosed to the public. If you need access to data, please contact Kun Zheng (ZhengK@cug.edu.cn) and Yan Liu (liuyan_@cug.edu.cn).

Author contributions. Kun Zheng was responsible for developing models and writing manuscripts; Yan Liu and Qiya Tan conducted model experiment and co-authored the manuscript; Jinbiao Zhang, Cong Luo, Siyu Tang Huihua Ruan, Yunlei Yi and Xiutao Ran were responsible for data screening and preprocessing.

Competing interests. The authors declare that they have no conflict of interest.

Acknowledgements. This research was funded by Science and Technology Planning Project of Guangdong Province, China
(No.2018B020207012).

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
