# Peer review of "GAN-argcPredNet v1.0: A Generative Adversarial Model for Radar Echo Extrapolation Based on Convolutional Recurrent Units"

_Geoscientific Model Development, 2021_

## Author Response (AR1)

**Reply**

1. There are many deep learning extrapolation methods developed for precipitation nowcasting, for example, ConvGRU, TrajGRU, PredRNN, etc. The authors are suggested to compare with some of these methods.

Thank you for your suggestions. One of the key points of this paper is to solve the problem of predictive image blur. Previous studies have shown the advantages of generative countermeasure network (GAN) in this problem. Your suggestion is great, we will add relevant experiments in the revision later.

We have supplemented the experiment in the experiment section

2. Some important references are missing.

Thank you for your reminder. The missing references you mentioned will be added when we submit the revision later.

We added quotes in the introduction section (lines 29 and 31)

3. Is it possible to provide a visualization of multi model image problem, or schematic, for the readers in the introduction?

Thanks for your question. We tried to draw, and finally found that the problem of multi-modal images is difficult to represent with visual images, but we added an explanation for this in the introduction.

We have modified the introduction (lines 41-44)

4. How did you arrive at the number of convolutional layers? Did you go through a period of architecture optimization? e.g. using hyper tune?

Thanks for your question. In order to get better discrimination effect, we optimized the network structure and performed a lot of training to compare and experiment. We tried a two-layer convolutional network for the first time, and found that the loss value of the discriminator soon stabilized, which to a certain extent reflects the weak discriminating effect, indicating that the network has room for improvement. In the end we got a 4-layer convolution discriminator, and many experiments show that it works best.

5. What is the role of data augmentation, if any, on the model performance? Is it possible to use synthetic data?

Thanks for your question. Are you talking about data augmentation? The paper did not use this method, which may be a direction of our

attention in the future.

6. The formatting of the equations seems to be wrong in places. Please check this e.g. in equation 1, the subscripts under argmax should be smaller than the main variables.

Thanks for your question. The expression of this formula means to extrapolate the next seven frames in the case of five inputs. The variables you mentioned can be understood as inputs.

Minor comments:

1. Abstract - 'The model composed..' should read the 'model is composed of the argcPredNet..' . In the proceeding line 'In generator, a gate control data memory and output are designed in the rgcLSTM prediction unit of the generator, thereby reducing the loss of spatiotemporal information' should read (i guess), 'In the generator, a gate controlling the memory and out are designed in the rgcLSTM component..'?

   We have made changes in lines 16-18

2. Page 2, line 42: 'which prefers to model unimodal distribution.' should

read 'which is better suited to a unimodel distribution'?

We made a modification on line 41

3. Relate work - this should be 'Related work'.

We made a modification on line 52

4. Page , line 57: please define GRU for the readers.

We added the definition on line 58

5. Page 3 '3.1 Model overview' Please change this to include the name of your model. Please also provide a few sentences telling the reader what each following section is describing.

We added some descriptions on line 82 and revised the title of 3.1

6. Page 3 line 86. 'WGAN-gp' please define what this is. expand the acronym for each first instance. Do bare in mind that readers should not need to know implicitly what a model acronym means.

We added a supplement on line 92

7. Page 3 line 91: 'Finally, use Adam to optimize the adversarial loss and then update the parameters of the discriminator, optimize the loss function of the' should read, Finally, we use the Adam optimiser for training..'

 We made a modification on line 97

8. I think your paper would really benefit from a table listing all model components that you bring together and a general algorithm workflow.

We added algorithm flow in 3.1, and tables in 3.2.2 and 3.3.1

9. Please add clarify in the figure 1 caption this is the new model [and name it]

We have modified the caption of Figure 1

All the lines mentioned refer to the revised version